# Fabrication of Cu Micromembrane as a Flexible Electrode

**DOI:** 10.3390/nano12213829

**Published:** 2022-10-29

**Authors:** Bo-Yao Sun, Wai-Hong Cheang, Shih-Cheng Chou, Jung-Chih Chiao, Pu-Wei Wu

**Affiliations:** 1Department of Materials Science and Engineering, National Yang Ming Chiao Tung University, Hsinchu 300, Taiwan; 2Department of Electrical and Computer Engineering, Southern Methodist University, Dallas, TX 75205, USA

**Keywords:** polypropylene micromembrane, polydopamine, polyethyleneimine, electroless Cu deposition, flexible electrode

## Abstract

A Cu micromembrane is successfully fabricated and validated as a porous flexible electrode. The Cu micromembrane is prepared by functionalizing individual polypropylene (PP) fibers in a polypropylene micromembrane (PPMM) using a mixture of polydopamine (PDA) and polyethyleneimine (PEI). The mixture of PDA and PEI provides adhesive, wetting, and reducing functionalities that facilitate subsequent Ag activation and Cu electroless plating. Scanning electron microscopy reveals conformal deposition of Cu on individual PP fibers. Porometer analysis indicates that the porous nature of PPMM is properly maintained. The Cu micromembrane demonstrates impressive electrical conductivities in both the X direction (1.04 ± 0.21 S/cm) and Z direction (2.99 ± 0.54 × 10^−3^ S/cm). In addition, its tensile strength and strain are better than those of pristine PPMM. The Cu micromembrane is flexible and mechanically robust enough to sustain 10,000 bending cycles with moderate deterioration. Thermogravimetric analysis shows a thermal stability of 400 °C and an effective Cu loading of 5.36 mg/cm^2^. Cyclic voltammetric measurements reveal that the Cu micromembrane has an electrochemical surface area of 277.8 cm^2^ in a 1 cm^2^ geometric area (a roughness factor of 227.81), a value that is 45 times greater than that of planar Cu foil.

## 1. Introduction

The development of flexible electrodes is of critical significance for possible applications in flexible electronics and energy storage devices [1,2]. Flexible electrodes can accommodate the larger mechanical stress incurred during device manufacturing and operation, and thus exhibit an improved lifetime compared with that of rigid counterparts. In the literature, the fabrication of flexible electrodes is often achieved by the metallization of polymeric substrates [3,4]. The metallization step typically involves physical vapor deposition or electroless deposition [5,6]. Between these two, electroless deposition is known to be an inexpensive route enabling better uniformity for the metal deposit. So far, the polymers chosen as the flexible substrate can be either compact solid (polytetrafluoroethylene or polyethylene terephthalate) [7,8] or porous solid (polyvinylidene difluoride or polyimide) [9,10]. It is recognized that a porous substrate allows for a larger surface area for chemical reaction and a greater loading of reactants. In addition, for any reaction involving the gaseous phase, the porous substrate becomes rather desirable for improved permeation and mass transport. 

Polymer substrates are often chemically inert, and therefore, an intimate bonding with the metal deposit is an important step in the fabrication of a robust flexible electrode. Previously in our laboratory, we have employed polydopamine (PDA) as an adhesive agent to facilitate electroless Cu deposition on a SiO_2_ substrate and the formation of dense RuO_2_ thin film on a glass slide [11,12]. PDA is the polymeric form of dopamine, which is a natural molecule found in mussels [13,14]. To date, PDA and its composites have been explored as surface functionalization agents to promote chemical bonding between heterogeneous surfaces [15]. This is because dopamine contains catechol and amine groups that are able to establish strong interactions with a wide variety of substrates via hydrophobic interaction, hydrogen bonds, and electrostatic attraction [16,17].

Cu is an inexpensive conductive metal that is used in household wiring and interconnects in semiconductor devices [18,19]. In addition, Cu has been adopted as a current collector for lithium-ion batteries and supercapacitors, and is used as an electrocatalyst for CO_2_ and nitrate reduction [20,21,22,23]. For electrochemical applications, it is desirable to have a large accessible area to engage the redox reaction. Therefore, it is necessary to develop a microporous Cu structure that serves as a reaction platform. We rationalize that by leveraging the adhesive and reducing abilities of PDA, the successful metallization of the polymer membrane can be achieved via electroless Cu deposition.

Recently, we selected polypropylene micromembrane (PPMM) as the polymeric substrate for its flexibility, porosity, and low cost. We demonstrated the fabrication of Ag-coated PPMM and Au/IrO_2_-coated PPMM, and we explored their possible applications in chlorine sensing and pH sensing [24,25]. In this work, we designed an effective fabrication scheme to fabricate a Cu micromembrane with impressive flexibility, mechanical strength, and electrical conductivity. Our processing steps entailed the use of PDA and polyethyleneimine (PEI) as surface functionalization agents that not only transformed the hydrophobic PP fibers into hydrophilic ones, but also served as reducing agents to facilitate the formation of Ag seedlings on the PP fibers for subsequent Cu electroless deposition. Comprehensive characterization was carried out to elucidate the physical, chemical, and mechanical properties of the flexible Cu micromembrane. 

## 2. Materials and Methods

### 2.1. Chemicals and Materials

Tris(hydroxymethyl)aminomethane hydrochloride (Tris-HCl), PEI (branched, MW 800 Da), chloramine-T hydrate, and phenol red sodium salt were purchased from Sigma-Aldrich (St. Louis, MO, USA). Dopamine hydrochloride (DA-HCl) was purchased from Acros Organic (Geel, Belgium). Potassium hydroxide, hydrochloride, sodium thiosulfate, and potassium bromide were purchased from SHOWA (Gunma, Japan). Ethylenediamine (En), anhydrous cobalt chloride, silver nitrite, and anhydrous copper chloride were purchased from Alfa Aesar (Tewksbury, MA, USA). Sodium chloride was purchased from Fluka (Munich, Germany). Sodium acetate was purchased from Vetec (St. Louis, MO, USA). All these chemicals were of analytical grade and were used without further purification. The PPMM (catalog number: 201PP-47-045-50; diameter of 47 mm; pore size of 450 nm; thickness of 200 µm) was purchased from Rone Scientific Inc. (New Taipei City, Taiwan).

### 2.2. Fabrication of Cu Micromembrane

The fabrication of the Cu micromembrane started with the functionalization of PP fibers in PPMM via the conformal deposition of PEI and PDA. The mixture of PEI and PDA contained 80 mg PEI and 100 mg DA-HCl (effective DA amount of 80 mg) in 50 mL of 50 mM Tris-HCl buffer solution. The pH of the mixture was adjusted to 8.5 by adding a minute amount of 0.1 M KOH aqueous solution. The PPMM in the PDA/PEI mixture was immersed under constant stirring for 20 h in the ambient atmosphere to allow the dissolved oxygen to polymerize the DA to PDA. Next, the functionalized PPMM underwent an activation step to form nucleation sites for subsequent electroless Cu deposition. The activation step entailed the immersion of functionalized PPMM in 40 mL of 0.1 M AgNO_3_ aqueous solution at 25 °C for 10 min so the PDA on the PP fibers was able to reduce the Ag^+^ ions for the formation of Ag colloids serving as the nucleation sites. In subsequent electroless Cu deposition, the sample was submerged in a Cu electroless bath that contained 0.05 M CuCl_2_, 0.15 M CoCl_2_, and 0.6 M En in deionized water. The pH for the plating bath was adjusted to 9.2 ± 0.1 by adding a minute amount of concentrated HCl. The En was used to complex both Co and Cu ions to form CoEn_3_^2+^ and CuEn_2_^2+^, respectively. The CoEn_3_^2+^ served as the reducing agent to reduce the CuEn_2_^2+^ to Cu, and the entire deposition step was performed under 300 rpm stirring at 25 °C for 30 min. Afterward, the sample was retrieved and washed by deionized water, followed by drying in N_2_ to minimize any parasitic Cu oxidation. The resulting Cu-plated functionalized PPMM was denoted as the Cu micromembrane. Figure 1 displays the processing steps involved in the fabrication of the Cu micromembrane. 

### 2.3. Materials Characterization

The morphologies of PPMM, functionalized PPMM, and Cu micromembrane were observed by a scanning electron microscope (SEM, JEOL JSM6700F). The X-ray diffraction (XRD) patterns for the PPMM and Cu micromembrane were obtained by a Bruker D2 Phaser with Cu K_α_ radiation (λ = 1.54 Å) as the X-ray source. Thermogravimetric analysis (TGA) was conducted using a Q900 thermogravimetric analyzer (TA instruments) to obtain the exact Cu loading in Cu micromembrane. The TGA experiment was conducted in a N_2_ atmosphere with a ramping rate of 5 °C/min from 25 to 700 °C. The mechanical properties of the PPMM and Cu micromembrane were studied with a microforce tester (Tytron 250 microforce testing system) to obtain the stress–strain profile. The sample was cut to 1 × 2 cm^2^, and was subjected to a horizontal force with a loading rate of 10 mm/min. The permeability and porosity of the PPMM and Cu micromembrane were recorded by a capillary flow porometer (PMI 1200) using Galwick (surface tension of 15.9 dynes/cm) as the wetting agent. The electrical resistance of the Cu micromembrane in both the *X*-axis and *Z*-axis directions was acquired by a two-point probe (Keithley 2400). A bending test was also performed in a custom-made device by which the Cu micromembrane was bent to 60° repeatedly for 10,000 cycles. The electrical resistance was measured as a function of bending cycles. The electrochemical surface area (ECSA) of Cu in the Cu micromembrane was determined in a three-electrode cell using Pt foil (2 × 2 cm^2^) and Ag/AgCl (3M) as the counter and reference electrodes, respectively. The electrolyte was 0.1 M NaHCO_3_ aqueous solution. The size of the working electrode was 1 × 1 cm^2^. The ECSA was determined by conducting cyclic voltammetric (CV) scans between 0.05 and 0.25 V (vs. Ag/AgCl) at scan rates of 10, 25, 50, 75, 100, 150, 200, 250, and 300 mV/s, separately. The potentiostat was a Versastat 4, and all electrochemical measurements were carried out at 25 °C. 

## 3. Results

### 3.1. Functionalization of PPMM

Figure 2a displays the molecular structure of PEI and PDA. As shown, PEI exhibits a branched structure with primary, secondary, and tertiary amino functional groups. In the literature, PEI is often used as a cross-linking agent because of its abundant amino groups [26]. On the other hand, DA contains chemical groups of amine, catechol, and phenyl, and thus shows strong affinity toward different materials via hydrogen bonding and electrostatic and hydrophobic interactions [17]. In addition, upon exposure to oxygen or oxidizing agents, DA is prone to be oxidized to form PDA. Figure 2b displays the chemical interaction that occurs between PEI and PDA. As shown, the amine group in PEI is able to covalently bond with the phenyl group in DA via the Schiff base or Michael addition reaction [15]. This relatively stronger covalent bonding reduces the self-polymerization of DA, producing a mixture of PEI and PDA as small aggregates uniformly deposited on the surface of PP fibers. 

Figure 2c displays the top-view SEM image of PP fibers in pristine PPMM. As shown, the PP fibers revealed a smooth surface, and their diameter was in the range of 0.9~6.5 μm. Because the arrangement of PP fibers is in a nonwoven state, the PPMM is expected to exhibit anisotropy in both mechanical properties and electrical conductivity. Figure 2d displays the top-view SEM image of PP fibers in functionalized PPMM. Apparently, after deposition of PEI/PDA, the PP fibers maintained their smooth surface morphology. In addition, from the inset of the high-magnification image, the PEI/PDA aggregates were well-dispersed, and their size was in the range of 5~15 nm. At this stage, the hydrophobic PPMM became hydrophilic, as evidenced by the quick wetting of a water droplet on the functionalized PPMM. 

### 3.2. Material Characterization of Cu Micromembrane

The realization of uniform Cu deposition on individual PP fibers of PPMM requires a proper seedling step that promotes subsequent electroless Cu deposition. Ag was chosen as the nucleation species because the catechol group of PDA is able to reduce Ag^+^ ions for the formation of Ag nanoparticles [24,27]. In our experience, without an activation step in AgNO_3_ aqueous solution, Cu is unable to form a uniform deposit on PP fibers. Instead, individual Cu nanoparticles separately form and scatter on the PP fibers. As a consequence, the resulting PPMM exhibited a large electrical resistivity, up to 2 × 10^5^ times greater than that of the counterpart undergoing an activation step. However, with a proper activation step, the Cu electroless deposition smoothly proceeded on individual PP fibers. 

Figure 3a displays the high-resolution SEM image of a PP fiber in the Cu micromembrane. Apparently, individual Cu granules between the sizes of 200 and 250 nm were in intimate contact with each other. Figure 3b displays the cross-sectional SEM image of PPMM that was cut in the middle to expose the PP fibers and their Cu deposits. As shown, the average thickness of the Cu deposits was around 725 nm. Figure 3c displays photographs showing the samples in different stages: pristine PPMM, functionalized PPMM, Ag-activated PPMM, and Cu micromembrane. 

Figure 4 displays the pore size distribution profiles for pristine PPMM, functionalized PPMM, and Cu micromembrane. The pore size and its distribution were defined by the following equations [28]:
(1)$$D=\frac{4\gamma cos\theta }{p}$$
(2)$$Q\left(p\right)=1-\frac{V_{wet}}{V_{dry}}$$
where *D* is the pore diameter, *γ* is the surface tension of the wetting agent, *cosθ* is the tortuosity factor of the wetting agent, *p* is the differential pressure, *Q*(*p*) is the cumulative distribution of pore size, *V*_wet_ is the flow rate for the sample with a wetting agent, and *V*_dry_ is the flow rate for the sample without a wetting agent. The largest pore size at *Q* = 1 is defined as the bubble point. On the other hand, the smallest pore size occurs at *Q* = 0 when *V*_wet_ = *V*_dry._ As shown in Figure 4, the pore size reduced, as expected, when the pristine PPMM underwent functionalization of PDA/PEI and further electroless Cu deposition. 

The permeability and pore size of a Cu micromembrane are important properties relevant to applications in membrane/filtering and electrochemistry. Table 1 provides the results for the pristine PPMM, functionalized PPMM, and Cu micromembrane. As listed, both pristine PPMM and functionalized PPMM demonstrated similar results in those parameters. Their minor differences were likely caused by variations in microstructure during the manufacturing of PPMM. In contrast, the pore size of the Cu micromembrane was considerably reduced compared with those of pristine and functionalized PPMM. This was consistent with the increased bubble point pressure. Notably, for a randomly distributed fibrous membrane, the same coating process is known to engender deposits with various thicknesses, and, thus, it is unreliable to exclusively define the coating thickness from the pore size measurements. For example, according to Wei et al., after the sputtering of Cu on nonwoven polypropylene fibers, the mean pore size was reduced, and the pore distribution became narrower [29]. In our study, the mean pore size of the Cu micromembrane decreased from 0.436 to 0.266 μm, and the standard deviation reduced from 2.92% to 2.08%.

Figure 5 displays the XRD patterns for the pristine PPMM, functionalized PPMM, and Cu micromembrane, as well as the standard JCPDS Cu (04-0836) and Ag (04-0783) for comparison purposes. For both pristine and functionalized PPMM, their XRD patterns demonstrated the amorphous nature of PPMM. In contrast, the Cu micromembrane exhibited notable diffraction signals consistent with those of fcc Cu. In addition, their intensities agreed well with what is expected from the Cu standard, suggesting its isotropic nature. The grain size of Cu was 29.7 nm as estimated from the Scherrer equation on the (220) plane. We did not record any diffraction signal from Ag, suggesting that the number of Ag nuclei was still insufficient for XRD detection. 

Figure 6 displays the TGA profiles for the pristine PPMM, functionalized PPMM, and Cu micromembrane. In our samples, the PP, PEI, and PDA were organic materials susceptible to thermal decomposition. As shown, for both pristine and functionalized PPMM, their TGA profiles were almost identical, in which a pyrolysis step was observed at 360 °C, and the decomposition was complete when the temperature surpassed 450 °C. At this stage, the remaining mass approached 0%, as only the nonvolatile carbonaceous residue was present. A slight difference in the residual weight was observed due to the presence of PEI and PDA in the functionalized PPMM. On the other hand, the Cu micromembrane revealed an improved thermal stability, in which the initial pyrolysis step occurred at 400 °C, followed by a weight loss between 400 and 470 °C that led to a residual mass of 41.3 wt %. This value indicated that the mass loading of Cu in the Cu micromembrane was 5.36 mg/cm^2^. 

Figure 7a displays the stress–strain curves for the pristine PPMM, functionalized PPMM, and Cu micromembrane. Apparently, these three samples exhibited similar patterns, in which an elastic deformation took place with a strain less than 1%, followed by a plastic deformation occurring with a strain at 98%, 101%, and 112%, respectively. Their corresponding tensile strength was 3.3, 3.5, and 3.7 MN/m^2^. We surmised that the limited elasticity among these three samples was caused by the nonwoven nature of PPMM, which constrained elongation along the direction of imposed stress. However, the relatively stronger tensile strength of the Cu micromembrane was attributed to the Cu deposit that formed a conformal coating on individual PP fibers. Interestingly, the functionalized PPMM was stronger than pristine PPMM, which we think was caused by the strong adhesive nature of PDA/PEI that increased the strength of the PP fibers. In addition, the deposition of PDA/PEI on pristine PPMM reduced its pore size, which also suggested a relatively larger solid loading for functionalized PPMM over that of pristine PPMM, rendering an improved mechanical strength. Figure 7b displays an enlarged view of the stress–strain curve in the elastic regime. As shown, the yield strength of pristine PPMM, functionalized PPMM, and Cu micromembrane was 0.9, 1.1, and 1.2 MN/m^2^, respectively. The improved yield strength of the Cu micromembrane was reasonably expected because the Cu deposit was mostly responsible for the load-bearing action. 

### 3.3. Electrical and Electrochemical Characterization of Cu Micromembrane

The electrical conductivity of the Cu micromembrane is not isotropic because the orientation of PP fibers is random along the X-Y plane (parallel to the PPMM surface) and not aligned in the Z direction (across the PPMM). Figure 8a illustrates the measurement setup for electrical resistance in both the X and Z directions. The electrical resistance was obtained by recording the voltage produced from the imposed current flow. To obtain the average and its standard deviation, 10 samples were measured. As expected, the Cu micromembrane exhibited anisotropic behavior, where the electrical resistance in the X and Z directions was 11.75 and 14.63 Ω, respectively. The electrical resistance was further converted to electrical conductivity (S/cm). For example, in the X direction, when the probe distance is much larger than the film thickness, the conversion equation is listed as followed.
(3)$$\sigma_{x}=1/\rho_{x}=\left(\frac{2n\lambda }{\pi t}\right)\times 1/R_{x}=\left(1/4.532\times t\right)\times 1/R_{x}$$
where *σ_x_* is the electrical conductivity, *ρ_x_* is the electrical resistivity in the X direction, *R_x_* is the measured resistance in the X direction, *λ* is the correction constant, and *t* is the thickness of the Cu micromembrane. For the Z direction, the conversion equation is listed as follows:(4)$$\sigma_{z}=1/\rho_{z}=t/\left(R_{z}\times A\right)$$
where *σ_z_* is the electrical conductivity in the Z direction, *ρ_z_* is the electrical resistivity in the Z direction, and *A* is the Cu micromembrane area for current flow (0.5 cm^2^).

Figure 8b displays a bar chart for the electrical resistivity in both the X and Z directions. Their corresponding electrical conductivity in the X and Z direction was 1.04 ± 0.21 and 2.99 ± 0.54 × 10^−3^ S/cm, respectively. Figure 8c illustrates the setup for the bending test. The Cu micromembrane was deformed to 60° at 0.5 mm/s and was subsequently released. The same action was repeated 10,000 times, and the electrical resistance was measured after completing the predetermined cycling number. Figure 8d displays the percentage of the variation in the electrical resistance in both the X and Z directions as a function of bending cycles. As shown, there was a 100% increase in electrical resistance for both the X and Y directions after 1000 cycles. After 10,000 bending cycles, the increase in electrical resistance for both the X and Z directions was raised to 327% and 565%, respectively. We also attempted to carry out linear fitting to determine the deterioration behavior of the Cu micromembrane. As shown in Figure 8d, the slope was 2.55 × 10^−2^ and 5.12 × 10^−2^ for the X and Z directions, respectively. Figure 8e displays the top-view SEM image of the Cu micromembrane after the bending test. Apparently, the Cu micromembrane retained similar characteristics to pristine PPMM, and the Cu deposit was strongly bonded to the PP without physical delamination. Figure 8f displays a photograph showing a Cu micromembrane that was bent, demonstrating its impressive flexibility.

The ECSA of Cu in the Cu micromembrane was determined by obtaining its double-layer capacitance during a CV scan in nonfaradic regime, and the capacitance was divided by the standard double-layer capacitance of planar Cu (27.5 µF/cm^2^) [30]. Figure 9a displays the CV profiles of the Cu micromembrane in 0.1 M NaHCO_3_ aqueous solution at scan rates of 10, 25, 50, 75, 100, 150, 200, 250, and 300 mV/s. The potential window of −0.75 and −0.55 V (vs Ag/AgCl) was selected because this range exhibited a negligible Faradaic reaction. As shown, the resulting CV curves exhibited a typical capacitive response that linearly increased with the scan rate. Figure 9b displays the corresponding CV profiles for the planar Cu foil used as the reference sample. Apparently, the planar Cu foil revealed a significantly reduced capacitive current response due to its limited geometric area. Figure 9c–d display the relation of differential current density (determined at −0.65 V (vs. Ag/AgCl)) as a function of scan rate. The double-layer capacitance of the Cu micromembrane and planar Cu foil was 7.64 and 0.169 mF/cm^2^, respectively. This indicated that the ECSA for the Cu micromembrane and planar Cu foil was 277.81 and 6.15 cm^2^ in a 1 cm^2^ geometric area, respectively. The roughness factor, defined as the ratio of the ECSA over the geometric area, was 227.81 and 6.15, respectively. This validated the porous nature of the Cu micromembrane, which rendered a large roughness factor. A linear slope was recorded for both the Cu micromembrane and planar Cu foil. This indicated that our samples were robust and that their surface in contact with the electrolyte was stable, meaning the ECSA measurement was reliable. 

## 4. Conclusions

We demonstrated the fabrication of a Cu micromembrane using a mixture of PDA and PEI as the adhesive, wetting, and reducing agents. Individual PP fibers in PPMM were coated with PDA/PEI, which was followed by the formation of Ag nuclei for subsequent electroless Cu deposition. The SEM images validated the formation of a conformal Cu overcoat on each PP fiber, and the XRD pattern indicated its crystalline nature. The Cu micromembrane exhibited impressive electrical conductivities in both the X and Z directions, and its porous nature was maintained. In addition, the Cu micromembrane revealed improved mechanical properties compared with those of pristine PPMM, and it was able to sustain 10,000 bending cycles with moderate loss of conductivity. Moreover, the Cu deposit was found to strongly bond to the PP fibers without physical delamination. The ECSA of the Cu micromembrane was investigated via a double-layer capacitor model, and it was found to be 45 times larger than that of planar Cu foil. This validated the porous nature of the Cu micromembrane.

## Figures and Tables

**Figure 1 nanomaterials-12-03829-f001:**
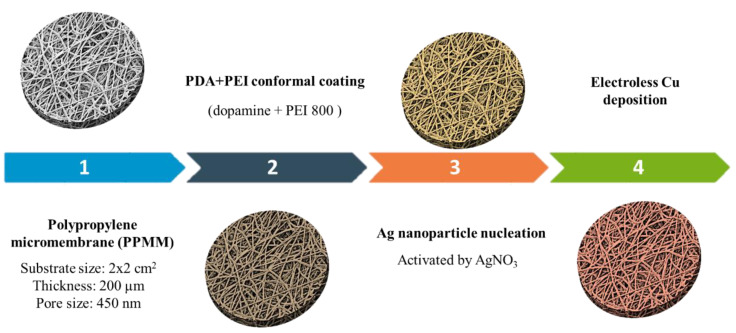
The schematic of processing steps involved in the fabrication of the Cu micromembrane.

**Figure 2 nanomaterials-12-03829-f002:**
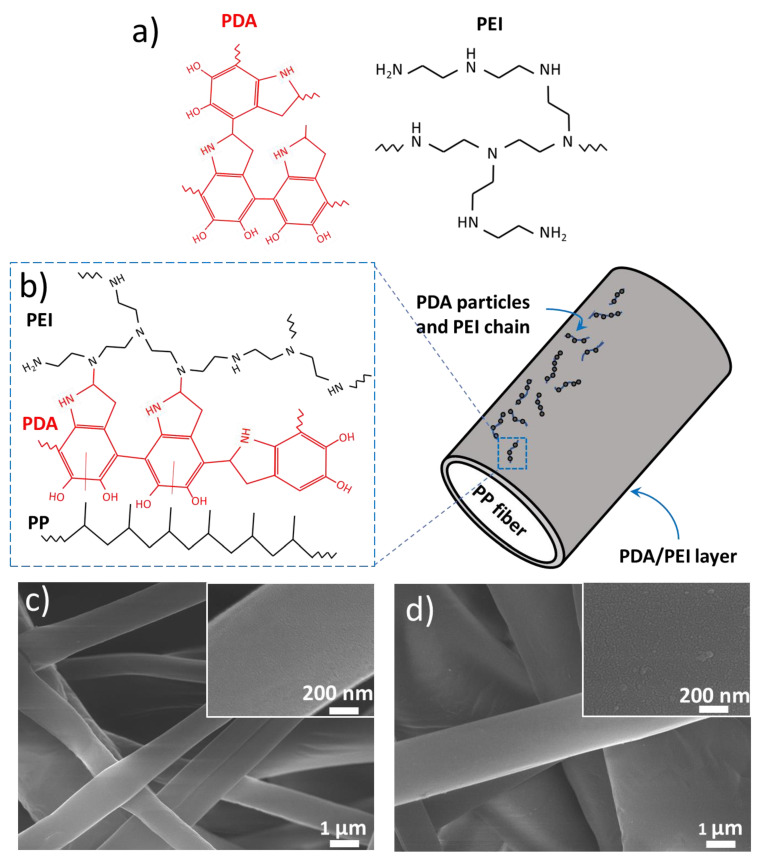
(**a**) The molecular structure of PEI and PDA. (**b**) Their chemical interaction to form PDA/PEI aggregates on the surface of PP fibers. (**c**) The top-view SEM image of PP fibers in pristine PPMM. (**d**) The top-view SEM image of PP fibers in functionalized PPMM. The insets are their respective high-magnification images.

**Figure 3 nanomaterials-12-03829-f003:**
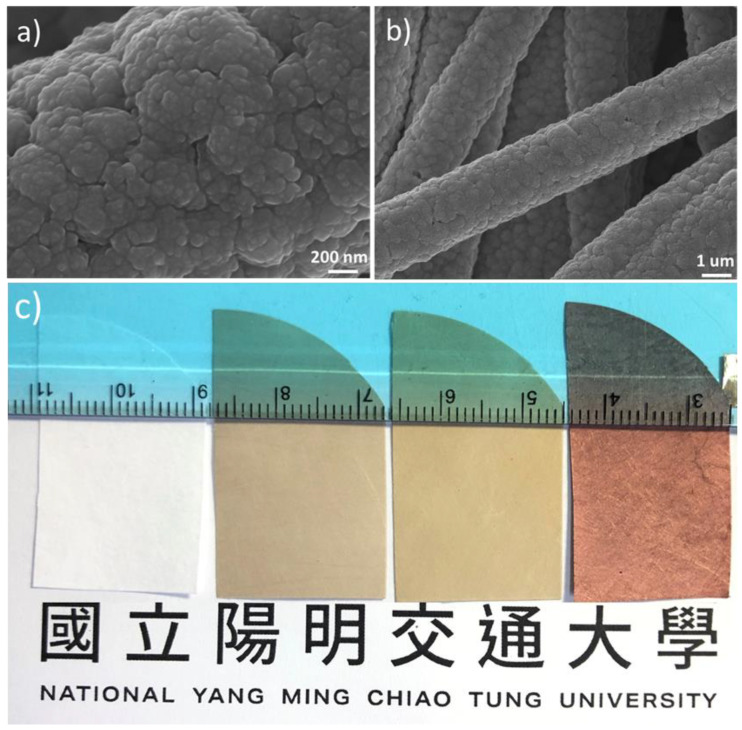
(**a**) The high-resolution SEM image of Cu micromembrane. (**b**) The cross-sectional SEM image of Cu micromembrane at low resolution. (**c**) The photograph of samples in different stages: pristine PPMM, functionalized PPMM, Ag-activated PPMM, and Cu micromembrane.

**Figure 4 nanomaterials-12-03829-f004:**
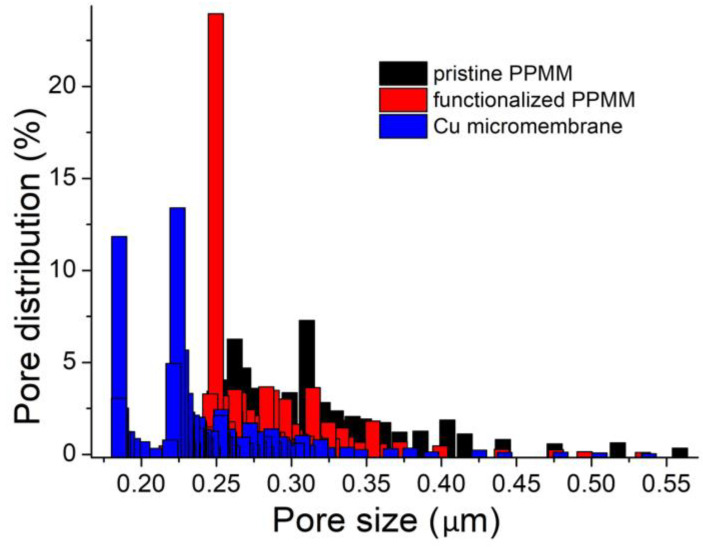
The pore size distribution for pristine PPMM, functionalized PPMM, and Cu micromembrane.

**Figure 5 nanomaterials-12-03829-f005:**
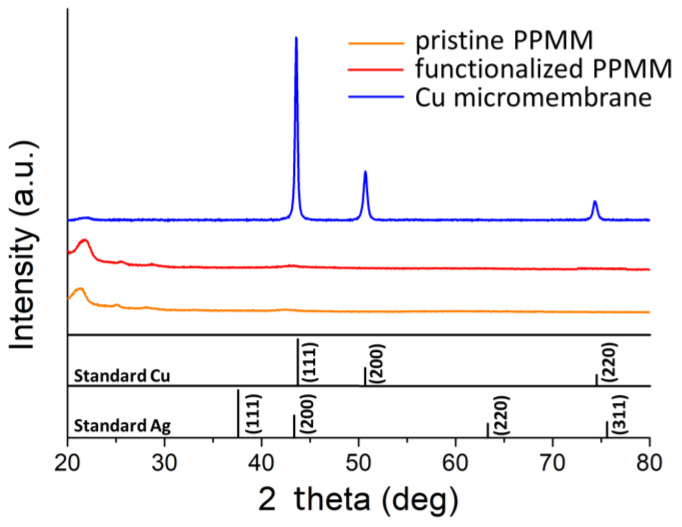
The XRD patterns for pristine PPMM, functionalized PPMM, and Cu micromembrane, as well as standard JCPDS of Cu (04-0836) and Ag (04-0783) for comparison purposes.

**Figure 6 nanomaterials-12-03829-f006:**
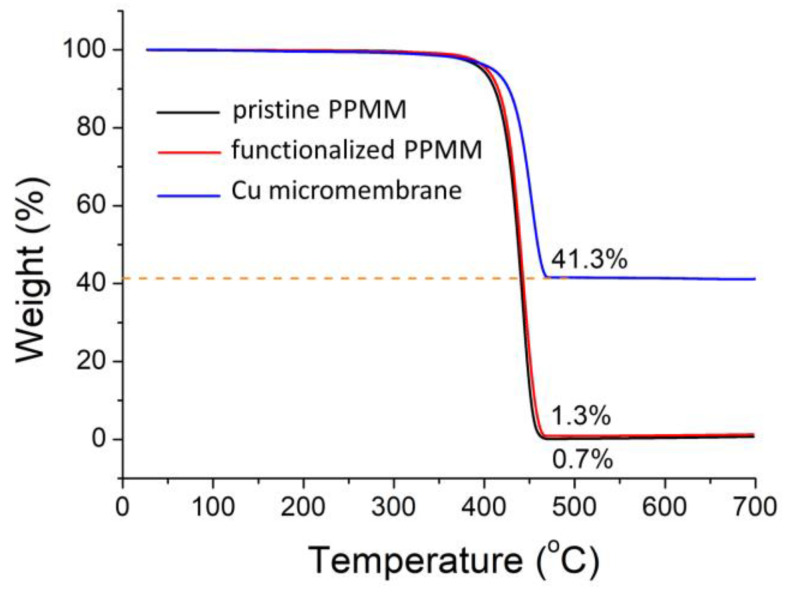
The TGA profiles for pristine PPMM, functionalized PPMM, and Cu micromembrane.

**Figure 7 nanomaterials-12-03829-f007:**
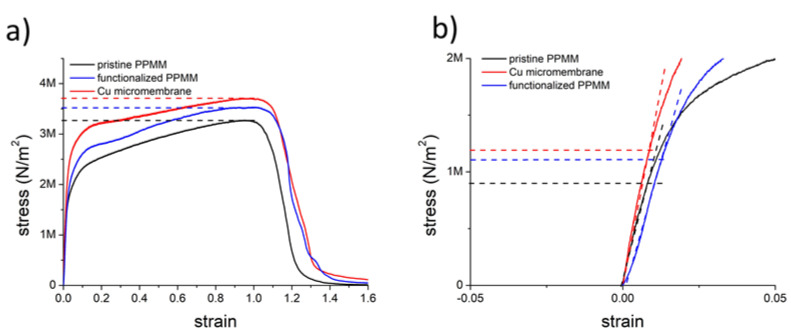
The stress–strain curve for pristine PPMM, functionalized PPMM, and Cu micromembrane: (**a**) overall and (**b**) enlarged profiles near the yield point.

**Figure 8 nanomaterials-12-03829-f008:**
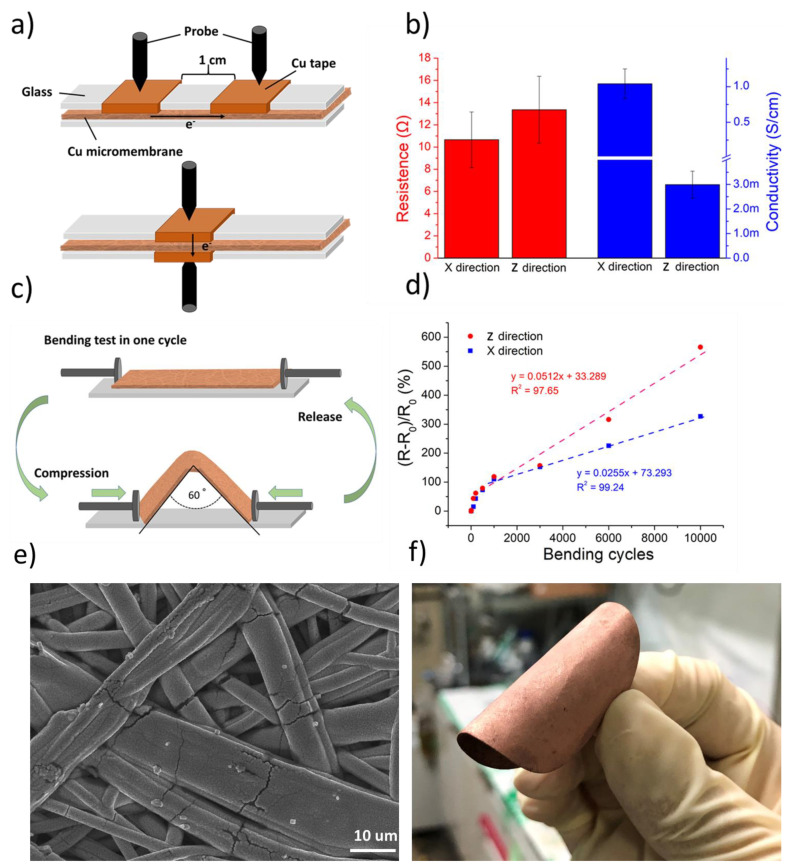
(**a**) The schematic of measurement setup for electrical resistance in both X and Z directions. (**b**) The bar chart for electrical resistance and conductivity in both X and Z directions. (**c**) The schematic of measurement setup for bending test. (**d**) The variation in percentage for electrical resistance as a function of bending cycles. (**e**) The top-view SEM images of Cu micromembrane after bending test. (**f**) A photograph showing the flexible nature of Cu micromembrane.

**Figure 9 nanomaterials-12-03829-f009:**
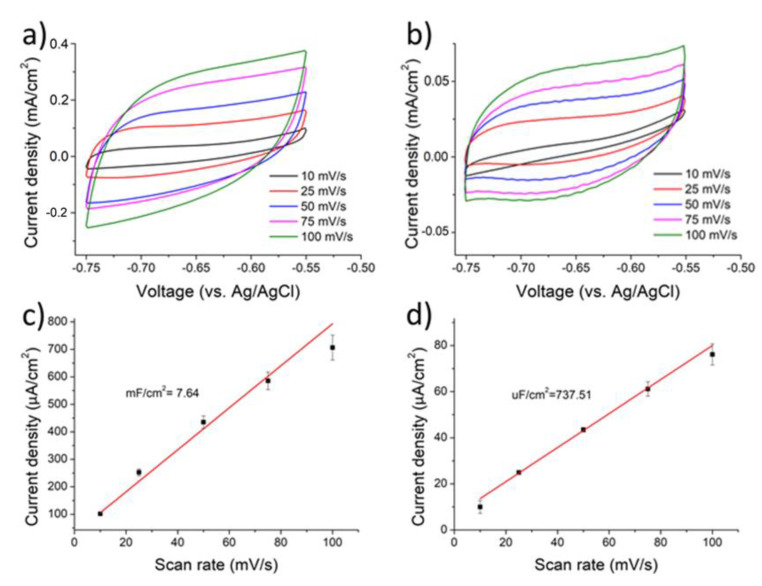
The CV scans in 0.1 M NaHCO_3_ aqueous solution with a potential window of −0.75 and −0.55 V (vs Ag/AgCl): (**a**) Cu micromembrane and (**b**) planar Cu foil. The difference in current density as a function of scan rate: (**c**) Cu micromembrane and (**d**) planar Cu foil.

**Table 1 nanomaterials-12-03829-t001:** Relevant parameters from capillary flow porometer for pristine PPMM, functionalized PPMM, and Cu micromembrane.

	PPMM		Cu Micromembrane
	Pristine	Functionalized	
Bubble point pore diameter (µm)	3.633	3.585	3.489
Bubble point pressure (psi)	1.816	1.84	1.891
Mean flow pore diameter (µm)	0.482	0.436	0.266
Mean flow pore pressure (psi)	13.691	15.136	24.77

## Data Availability

Not applicable.

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
