# Peer review of "Fabrication of Cu Micromembrane as a Flexible Electrode"

_nanomaterials, 2022, doi:10.3390/nano12213829_

Round 1

Reviewer 1 Report

The manuscript having the title Fabrication of Cu Micromembrane as a Flexible  Electrode is clearly written and represents an improved approach, by extending the number of components into the membrane (polydopamine (PDA) and polyethyleneimine (PEI), thus resulting better flexibility, mechanical strength, and electrical conductivity than in bare polypropylene micromembrane. The purpose of the work is appropriate to recent research in the field and is clearly presented. 

The reported work is based on very well-selected actual references and the the fabrication of Cu micromembrane is described in detail.

Some shortcomings are to be solved:

1. Please reformulate " substrate could be either solid (polytetrafluoroethylene, polyethylene terephthalate) [7,8] or porous  (polyvinylidene difluoride, polyimide) 

A solid could be porous, maybe compact or porous solid

2.  Did you experimented the use of colloidal Cu alone electroless deposition?

3. In the Chapter RESULTS, please insert some sub-chapters in order to better be understandable the flow of the work.

4. Please introduce "The electrochemical surface area"  before (ECSA) abbreviation and comment the significance of linear plots  in Figure 9.

5. In CONCLUSIONS section, please find an alternative for avoiding "In addition" repetition.

Reviewer 2 Report

 A Cu micromembrane is fabricated, and the conductivity and flexibility were verified. Eventually, the property of the micromembrane was exploded as a flexible electrode. The design of the experiment and results were clearly presented. Sufficient results can support the conclusion. I would like to recommend it for publication after considering the following comments:

1. The authors claimed that functionalization has no effect on the pore size. However, from Figure 4, functionalized vs Pristine, it seems like functionalized sample has smaller pores. Significantly, there is a high distribution peak at 0.25 um. Even though the numbers from table 1 show that there is only a small difference between them. However, it still suggested a smaller pore size. The effect of functionalization may need further verification. 

The stress-strain curve didn’t include the functionalized sample. However, the effect of fictionalization should be considered. Otherwise, it raises doubts about whether the improvement in tensile strength is solely related to the Cu deposit. 

The SEM image in Figure 8e does not have the same magnification compared to previous SEM images. Thus, it is not very clear to compare them. 

In Figures, 9a and 9b, the unit of the Y axis should be the same, for easier comparison. Otherwise, it is not straightforward. 
